# Complete Neural Networks
# for Complete Euclidean Graphs

## Abstract

Neural networks for point clouds, which respect their natural invariance to per-
mutation and rigid motion, have enjoyed recent success in modeling geometric
phenomena, from molecular dynamics Reiser et al. [2022] to recommender systems
Yi et al. [2023]. Yet, to date, no architecture with polynomial complexity is known
to be *complete*, that is, able to distinguish between any pair of non-isomorphic
point clouds. We fill this theoretical gap by showing that point clouds can be
completely determined, up to permutation and rigid motion, by applying the 3-WL
graph isomorphism test to the point cloud's centralized Gram matrix. Moreover, we
formulate a Euclidean variant of the 2-WL test and show that it is also sufficient to
achieve completeness. We then show how our complete Euclidean WL tests can be
simulated by a Euclidean graph neural network of moderate size and demonstrate
their separation capability on highly-symmetrical point clouds.

## 1 Introduction

A point cloud is a collection of $n$ points in $\mathbb{R}^d$, where typically in applications $d = 3$. Machine
learning on point clouds is an important task with applications in chemistry Gilmer et al. [2017],
Wang et al. [2022], physical systems Finzi et al. [2021] and image processing Ma et al. [2023]. Many
successful architectures for point clouds are invariant by construction to the natural symmetries of
point clouds: permutations and rigid motions.

The rapidly increasing literature on point-cloud networks with permutation and rigid-motion sym-
metries has motivated research aimed at theoretically understanding the expressive power of the
various architectures. This analysis typically focuses on two closely related concepts: *Separation*
and *Universality*. We say an invariant architecture is *separating*, or *complete*, if it can assign distinct
values to any pair of point clouds that are not related by symmetry. An invariant architecture is
*universal* if it can approximate all continuous invariant functions on compact sets. Generally speaking,
these two concepts are essentially equivalent, as discussed in Villar et al. [2021], Joshi et al. [2022],
Chen et al. [2019], and in our context, in Appendix A.

Dym and Maron [2020] proved that the well-known Tensor Field Network Thomas et al. [2018]
invariant architecture is universal, but the construction in their proof requires arbitrarily high-order
representations of the rotation group. Similarly, universality can be obtained using high-order
representations of the permutation group Lim et al. [2022]. However, prior to this work, it was not
known whether the same theoretical guarantees can be achieved by realistic point-cloud architectures
that use low-dimensional representations, and whose complexity has a mild polynomial dependency
on the data dimension. In the words of Pozdnyakov and Ceriotti [2022]: *"...provably universal
equivariant frameworks are such in the limit in which they generate high-order correlations... It is
an interesting, and open, question whether a given order suffices to guarantee complete resolving
power."* (p. 6). We note that it is known that separation of point clouds in polynomial time in $n$

is possible, assuming that $d$ is fixed (e.g., $d = 3$) Arvind and Rattan [2014], Dym and Kovalsky [2019], Kurlin [2022]. What still remains to be established is whether separation is achievable for common invariant machine learning models, and more generally, whether separation can be achieved by computing a continuous invariant feature that is piecewise differentiable.

In this paper, we give what seems to be the first positive answer to this question. We focus on analyzing a popular method for the construction of invariant point-cloud networks via *Graph Neural Networks (GNNs)*. This is done in two steps: first, point clouds are represented as a *Euclidean graph-* which we define to be a complete weighted graph whose edge features are simple, rotation-invariant features: the inner products between pairs of (centralized) points. We then apply permutation-invariant *Graph Neural Networks (GNNs)* to the Euclidean graphs to obtain a rotation- and permutation-invariant global point-cloud feature. This leads to a rich family of invariant point-cloud architectures, which is determined by the type of GNN chosen.

The most straightforward implementation of this idea would be to apply the popular message passing GNNs to the Euclidean graphs. One could also consider applying more expressive GNNs. For combinatorial graphs, it is known that message-passing GNNs are only as expressive as the 1-WL graph isomorphism test. There exists a hierarchy of $k$-WL graph isomorphism tests, where larger values of $k$ correspond to more expressive, and more expensive, graph isomorphism tests. There are also corresponding GNNs that simulate the $k$-WL tests and have an equivalent separation power Morris et al. [2018], Maron et al. [2019]. One could then consider applying these more expressive architectures to Euclidean graphs, as suggested in Lim et al. [2022]. Accordingly, we aim to answer the following questions:

**Question 1** For which $k$ is the $k$-WL test, when applied to Euclidean graphs, complete?

**Question 2** Can this test be implemented in polynomial time by a continuous, piecewise-differentiable architecture?

We begin by addressing Question 1. First, we consider a variation of the WL-test adapted for point clouds, which we refer to as 1-*EWL* ('E' for Euclidean). This test was first proposed by Pozdnyakov and Ceriotti [2022], where it was shown that it cannot distinguish between all 3-dimensional point clouds, and consequently, neither can GNNs like Victor Garcia Satorras [2021], Schütt et al. [2017], which can be shown to simulate it. Our first result, described in Section 2.1, balances this by showing that two iterations of 1-EWL are enough to separate *almost any* pair of point clouds.

To achieve complete separationfor *all* point clouds, we consider higher-order $k$-EWL tests. We first consider a natural adaptation of $k$-WL for Euclidean graphs, which we name the *Vanilla-$k$-EWL* test. In this test, the standard $k$-WL is applied to the Euclidean graph induced by the point clouds. We show that when $k = 3$, this test is complete for 3-dimensional point clouds. Additionally, we propose a variant of the Vanilla 2-EWL, which incorporates additional geometric information while having the same complexity. We call this test the 2-*EWL* test, and show that it is complete on 3D point clouds. We also propose a natural variation of 2-EWL called 2-*SEWL*, which can distinguish between point clouds that are related by a reflection. This ability is important for chemical applications, as most biological molecules that are related by a reflection are *not* chemically identical Kapon et al. [2021] (this molecular property is called *chirality*).

We next address the second question of how to construct a GNN for Euclidean data with the same separation power as that of the various $k$-EWL tests we describe. For combinatorial graphs, such equivalence results rely on injective functions defined on multisets of discrete features Xu et al. [2018]. For Euclidean graphs, one can similarly rely on injective functions for multisets with continuous features, such as those proposed in Dym and Gortler [2023]. However, a naive application of this approach leads to a very large number of hidden features, which grows exponentially with the number of message-passing iterations (see Figure 2). We show how this problem can be remedied, so that the number of features needed depends only linearly on the number of message-passing iterations.

To summarize, our main results in this paper are:

1. We show that two iterations of 1-EWL can separate *almost all* point clouds in any dimension.

2. We prove the completeness of a single iteration of the vanilla 3-EWL for point clouds in $\mathbb{R}^3$.

3. We formulate the 2-SEWL and 2-EWL tests, and prove their completeness for point clouds in $\mathbb{R}^3$.

4. We explain how to build differentiable architectures for point clouds with the same separation power as Euclidean $k$-WL tests, with reasonable complexity.

**Experiments** In Section 5 we present synthetic experiments that demonstrate that 2-SEWL can separate challenging point-cloud pairs that cannot be separated by several popular architectures.

**Disambiguation: Euclidean Graphs** In this paper we use a simple definition of a Euclidean graph as the centralized Gram matrix of a point cloud, and focus on a fundamental theoretical question related to this representation. In the learning literature, terms like 'geometric graphs' (not used here) could refer to graphs that have both geometric and non-geometric edge and vertex features, or graphs where pairwise distances are only available for specific point pairs (edges in an incomplete graph).

## 1.1 Related Work

**Euclidean WL** Pozdnyakov and Ceriotti [2022] showed that 1-EWL is incomplete for 3-dimensional point clouds. Joshi et al. [2022] defines separation for a more general definition of geometric graph, which combines geometric and combinatorial features. This work holds various interesting insights for this more general problem but they do not prove completeness as we do here.

**Other complete constructions** As mentioned earlier, Dym and Maron [2020] proved universality with respect to permutations and rigid motions for architectures using high-dimensional representations of the rotation group. Similar results were obtained inFinkelshtein et al. [2022], Gasteiger et al. [2021]. In Lim et al. [2022] universality was proven for Euclidean GNNs with very high-order permutation representations. In the planar case $d = 2$, universality using low-dimensional features was achieved in Bökman et al. [2022]. For $d \geq 3$ our construction seems to be the first to achieve universality using low dimensional representations.

For general fixed $d$, there do exist algorithms that can separate point clouds up to equivalence in polynomial time, but they do not seem to lend themselves directly to neural architectures. In Kurlin [2022], Widdowson and Kurlin [2023] complete tests are described, but they represent each point cloud as a 'multiset of multisets' rather than as a vector as we do, and so are not suitable for gradient descent based learning. Efficient tests for equivalence of Euclidean graphs were described in Brass and Knauer [2000], Arvind and Rattan [2014], but they compute features that do not depend continuously on the point cloud.

**Weaker notions of universality** In Widdowson and Kurlin [2022] the authors suggest a method for distinguishing almost every point clouds up to equivalence, similar to our result here on 1-EWL. Similarly, efficient separation/universality can also be obtained for point clouds with distinct principal axes Puny et al. [2021], Kurlin [2022]. Another setting in which universality is easier to obtain is when only rigid symmetries are considered and permutation symmetries are ignored Wang et al. [2022], Villar et al. [2021], Victor Garcia Satorras [2021]. All these results do not provide universality for *all* point clouds, with respect to the joint action of permutations and rigid motions.

## Mathematical notation

A (finite) *multiset* $\{\!\{y_1, \ldots, y_N\}\!\}$ is an unordered collection of elements where repetitions are allowed.

Let $\mathcal{G}$ be a group acting on a set $\mathcal{X}$. For $X, Y \in \mathcal{X}$, we say that $X \underset{\mathcal{G}}{=} Y$ if $Y = gX$ for some $g \in \mathcal{G}$.

We say that a function $f : \mathcal{X} \to \mathcal{Y}$ is *invariant* if $f(gx) = f(x)$ for all $x \in X, g \in G$. We say that $f$ is *equivariant* if $\mathcal{Y}$ is also endowed with some action of $G$ and $f(gx) = gf(x)$ for all $x \in \mathcal{X}, g \in \mathcal{G}$. A separating invariant mapping is an invariant mapping that is injective, up to group equivalence:

**Definition 1.1** (Separating Invariant)**.** Let $\mathcal{G}$ be a group acting on a set $\mathcal{X}$. We say $F : \mathcal{X} \to \mathbb{R}^K$ is a $\mathcal{G}$-*separating invariant* with *embedding dimension* $K$ if for all $X, Y \in \mathcal{X}$, $F(X) = F(Y) \Leftrightarrow X \underset{\mathcal{G}}{=} Y$.

We focus on the case where $\mathcal{X}$ is some Euclidean domain. To enable gradient-based learning, we shall need separating mappings that are continuous everywhere and differentiable almost everywhere.

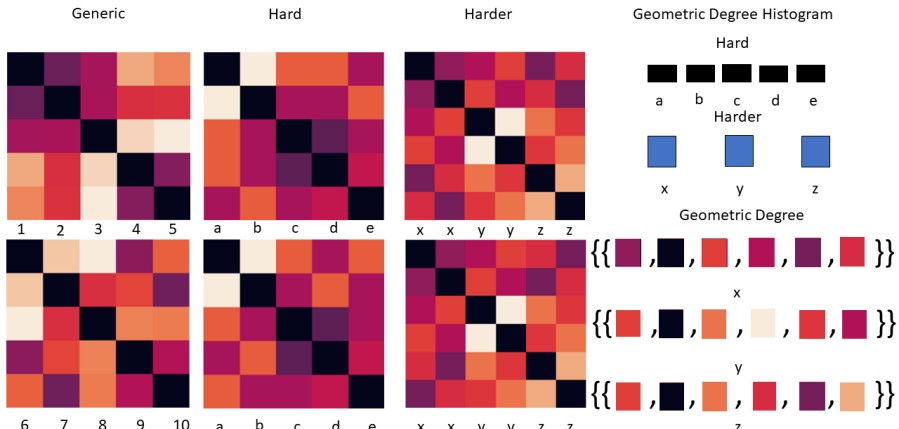

Figure 1: Distance matrices (Left), geometric degree histogram (Right) of pairs of point clouds. The *generic* pair is a randomly sampled pair of point clouds. Notice each of the nodes in each of the clouds has a distinct geometric degree. The *Hard* pair exhibits a distinct geometric degree for each node, but only within each point cloud, that is the pair shares an identical geometric degree histogram. The *Harder* example is a pair of point clouds with identical geometric degree histogram, and each point cloud is comprised of three pairs of points, with each pair having an identical geometric degree. Examples from Pozdnyakov and Ceriotti [2022] and Pozdnyakov et al. [2020].

The symmetry group we consider for point clouds $(x_1, \ldots, x_n) \in \mathbb{R}^{d \times n}$ is generated by a rotation matrix $R \in \mathcal{SO}(d)$, and a permutation $\sigma \in S_n$. These act on a point cloud by

$$(R, \sigma)_*(x_1, \ldots, x_n) = (Rx_{\sigma^{-1}(1)}, \ldots, Rx_{\sigma^{-1}(n)}).$$

We denote this group by $\mathcal{SO}[d, n]$. In some instances, reflections $R \in \mathcal{O}(d)$ are also permitted, leading to a slightly larger symmetry group, which we denote by $\mathcal{O}[d, n]$. Our goal shall be to construct separating invariants for these groups. For the sake of brevity, we do not discuss *translation* invariance and separation, as these can easily be achieved by centering the input point clouds, once $\mathcal{SO}[d, n]$ (or $\mathcal{O}[d, n]$) separating invariants are constructed, see Dym and Gortler [2023].

For simplicity of notation, throughout this paper, we focus on the case $d = 3$. In Appendix C we explain how our constructions and theorems can be generalized to $d > 3$.

## 2 Euclidean Graph Isomorphism Tests

The $k$-WL Graph Isomorphism Test Weisfeiler and Leman [1968] is a classical paradigm for testing the isomorphism of combinatorial graphs, which we shall now briefly describe. Let $\mathcal{G}$ be a graph with vertices indexed by $[n] = \{1, 2, \ldots, n\}$. We denote each ordered $k$-tuple of vertices by a multi-index $\mathbf{i} = (i_1, \ldots, i_k) \in [n]^k$. Essentially, for each such $k$-tuple $\mathbf{i}$, the test maintains a *coloring* $\mathbf{C}(\mathbf{i})$ that belongs to a discrete set, and updates it iteratively. First, the coloring of each $k$-tuple is assigned an initial value that encodes the *isomorphism type* of the corresponding $k$-dimensional subgraph:

$$\mathbf{C_{(0)}} = \mathbf{C_{(0)}}(\mathbf{i}), \ \mathbf{i} \in [n]^k. \tag{1}$$

Then the color of each $k$-tuple $\mathbf{i}$ is iteratively refined according to the colors of its 'neighboring' $k$-tuples. The update rule is given by

$$\mathbf{C_{(t+1)}}(\mathbf{i}) = \mathbf{Embed}^{(t+1)} \left( \mathbf{C_{(t)}}(\mathbf{i}), \{\!\!\{ \left( \mathbf{C_{(t)}}(\mathbf{i}[j \setminus 1]), \ldots, \mathbf{C_{(t)}}(\mathbf{i}[j \setminus k]) \right) \mid j \in [n] \}\!\!\} \right), \tag{2}$$

where $\mathbf{i}[j \setminus t]$ is the multi-index $\mathbf{i}$ with its $t$-th coordinate replaced by $j$; e.g. for $j = 1$, $\mathbf{i}[j \setminus 1] = (j, i_2, \ldots, i_k)$. **Embed** is a function that maps its input injectively to some discrete set. This process is repeated $T$ times to obtain a final coloring $\{\!\!\{ \mathbf{C_{(T)}}(\mathbf{i}) \}\!\!\}_{\mathbf{i} \in [n]^k}$. A global label is then calculated by

$$\mathbf{C_{\mathcal{G}}} = \mathbf{Embed}^{(T+1)} \left( \{\!\!\{ \mathbf{C_{(T)}}(\mathbf{i}) \mid \mathbf{i} \in [n]^k \}\!\!\} \right),$$

where **Embed**$^{(T+1)}$ is a function that maps label-multisets injectively to some discrete set.

To test whether two graphs $\mathcal{G}$ and $\mathcal{G}'$ are isomorphic, the $k$-WL test computes the corresponding colorings $\mathbf{C}_\mathcal{G}$ and $\mathbf{C}_{\mathcal{G}'}$ for some chosen $T$. If $\mathbf{C}_\mathcal{G} \neq \mathbf{C}_{\mathcal{G}'}$ then $\mathcal{G}$ and $\mathcal{G}'$ are guaranteed not to be isomorphic, whereas if $\mathbf{C}_\mathcal{G} = \mathbf{C}_{\mathcal{G}'}$, then $\mathcal{G}$ and $\mathcal{G}'$ may either be isomorphic or not, and the test does not, in general, provide a decisive answer for combinatorial graphs. It is known that this test is able to distinguish a strictly larger class of combinatorial graphs for every strict increase in the value of k, i.e. it is a strict hierarchy of tests in terms of distinguishing power Cai et al. [1992], Grohe [2017].

**Vanilla-$k$-WL tests** As a first step from a combinatorial to a Euclidean setting, we identify each point cloud $X = (x_1, \ldots, x_n) \in \mathbb{R}^{d \times n}$ with a complete graph on $n$ vertices, wherein each edge $(i, j)$ is endowed with the weight $w_{ij}(X) = \langle x_i, x_j \rangle$. We name such a graph a *Euclidean graph*. Similarly to $k$-WL for combinatorial graphs, $k$-WL for Euclidean graphs maintains a coloring of the $k$-tuples of vertices. However, the initial color of each $k$-tuple $\mathbf{i}$ is not a discrete label as in the combinatorial case, but rather a $k \times k$ matrix of continuous features, which represent all edge weights $w_{ij}$ corresponding to pairs of indices from $\mathbf{i}$. We will call the $k$-WL test defined by this initial coloring the **vanilla $k$-WL** test. This test is invariant by construction to reflections, rotations, and permutations. We note that our definition of the vanilla $k$-EWL test via inner products follows that of Lim et al. [2022]. Another popular, and essentially equivalent, formulation, uses distances instead.

**$k$-EWL tests** An inherent limitation of the Vanilla-1-EWL test is that no pairwise Euclidean information is passed, yielding it rather uninformative. Indeed, Pozdnyakov and Ceriotti [2022] proposed a Euclidean analog of the 1-WL test, where the update rule (2) is replaced with

$$\mathbf{C}_{(\mathbf{t+1})}(i) = \mathbf{Embed}^{(\mathbf{t})}\left(\mathbf{C}_{(\mathbf{t})}(i), \{\!\{\left(\mathbf{C}_{(\mathbf{t})}(j), \|x_i - x_j\|\right), j \neq i\}\!\}\right). \tag{3}$$

We call this test the 1-**EWL** test. This formulation is motivated by the fact that many symmetry-preserving networks for point clouds are in fact a realization of it, though they use **Embed** functions that are continuous and, in general, may assign the same value to different multisets. Consequently, the separation power of these architectures is at most that of 1-EWL with discrete injective hash functions. Moreover, the separation power will be equivalent if continuous injective multiset functions are used for embedding, as we discuss in Section 4.

The 1-EWL test strengthens the Vanilla-1-EWL test by allowing the messages passed to a node in each step to contain not only previous colorings but also geometric information in the form of pairwise distances. More generally, we shall use the term $k$-**EWL** to refer to tests that follow the Euclidean $k$-WL paradigm, but incorporate geometric invariants into the message-passing procedure. In particular, for point clouds with dimension 3, we define the 2-SEWL test ('SE' for *Special Euclidean*) by replacing the update step (2) with

$$\mathbf{C}_{(\mathbf{t+1})}(i, j) = \mathbf{Embed}^{(t)}\left(\mathbf{C}_{(\mathbf{t})}(i, j), \{\!\{\left(\mathbf{C}_{(\mathbf{t})}(k, j), \mathbf{C}_{(\mathbf{t})}(i, k), \langle x_i \times x_j, x_k \rangle\right)\}\!\}_{k=1}^n\right). \tag{4}$$

Note that $\langle x_i \times x_j, x_k \rangle$ is equal to the determinant of the $3 \times 3$ matrix whose rows are the three vectors $x_i, x_j, x_k$, which makes this a natural choice as all polynomial invariants of $\mathcal{SO}(3)$ are generated by these determinants and the inner products we use for the initial coloring Kraft and Procesi [1996].

We note that, Using the fact that $O(3)$ is just two copies of $SO(3)$, it is not difficult to generalize 2-SEWL to a complete $\mathcal{O}[3, n]$ test, which we name 2-EWL. for general $d$, similar complete $(d-1)$-SEWL and $(d-1)$-EWL tests can be formulated for point clouds in $\mathbb{R}^d$ via the Hodge-star operator; see Appendix C for more details.

In the rest of this section, we shall prove that the 2-SEWL, 2-EWL and vanilla 3-EWL tests are complete when applied to $\mathbb{R}^{3 \times n}$, even when using a single iteration ($T = 1$). We shall also show that two iterations of the 1-EWL test is complete, except on a set of measure zero.

## 2.1 Generic completeness of 1-EWL

The separation power of 1-EWL is closely linked to the notion of *geometric degree*: For a point cloud $X = (x_1, \ldots, x_n)$, we define the geometric degree $d(i, X)$ of the $i$th point, and the induced geometric degree histogram $d_H(X)$, to be the multisets

$$d(i, X) = \{\!\{\|x_1 - x_i\|, \ldots, \|x_n - x_i\|\}\!\}, \quad d_H(X) = \{\!\{d(1, X), \ldots, d(n, X)\}\!\}.$$

It is not difficult to see that if $d_H(X) \neq d_H(Y)$ then $X$ and $Y$ can be separated by a single 1-EWL iteration . An example of such a pair is shown in the left of Figure 1. With two 1-EWL iterations, we show that can separate $X$ and $Y$ even if $d_H(X) = d_H(Y)$, provided that they both belong to the set of point clouds defined by

$$\mathbb{R}^{3 \times n}_{distinct} = \{X \in \mathbb{R}^{3 \times n} | \, d(i, X) \neq d(j, X) \ \forall i \neq j\}.$$

Such an example, taken from Pozdnyakov et al. [2020], is visualized in the middle column of Figure 1

**Theorem 2.1.** *Two iterations of the* 1*-EWL test assign two point clouds* $\mathcal{X}, Y \in \mathbb{R}^{3 \times n}_{distinct}$ *the same value, if and only if* $X \underset{\mathcal{O}[3,n]}{=} Y$.

In the appendix we show that the complement of $\mathbb{R}^{3 \times n}_{distinct}$ has measure zero. Thus this result complements long-standing results for combinatorial graphs, stating that 1-WL can classify almost all such graphs as the number of nodes tends to infinity Babai et al. [1980].

The right-most pair of point clouds ('Harder') in Figure 1 is taken from Pozdnyakov and Ceriotti [2022]. The degree histograms of these point clouds are identical, and they are not in $\mathbb{R}^{3 \times n}_{distinct}$. Pozdnyakov and Ceriotti [2022] show that this pair cannot be separated by any number of 1-EWL iterations.

## 2.2 Is 1-EWL All You Need?

Theorem 2.1 shows that the probability of a failure of the 1-EWL is zero. A natural question to ask is whether more powerful tests are needed. We believe the answer to this question is yes. Typical hypothesis classes used for machine learning, such as neural networks, are Lipschitz continuous Gama et al. [2020]. In this setting, failure to separate on a measure zero set could have implications for non-trivial positive measure. This phenomenon is depicted in the figure in the inset. On the right, a plot of a Gaussian distribution centered at $x \in \mathbb{R}$, depicting a target function is shown in blue. In red, a schematic plot of how a Lipschitz continuous function that does not distinguish $x$ from $y$ would model the target function.

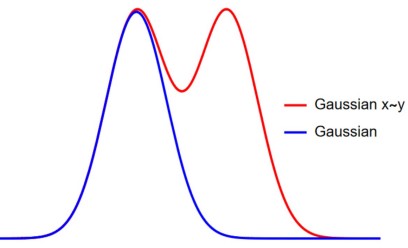

## 3  2-SEWL and Vanilla 3-EWL are complete

We now prove that the vanilla 3-EWL test is complete.

**Theorem 3.1.** *For every* $X, Y \in \mathbb{R}^{3 \times n}$, *a single iteration of the vanilla* 3*-EWL test assigns* $X$ *and* $Y$ *the same value if and only if* $X \underset{\mathcal{O}[3,n]}{=} Y$.

*Proof.* First, it is clear that if $X \underset{\mathcal{O}[3,n]}{=} Y$ then $\mathbf{C}_{\mathcal{G}}(X) = \mathbf{C}_{\mathcal{G}}(Y)$ since the vanilla 3-EWL test is invariant by construction. The challenge is proving the other direction. To this end, let us assume that $\mathbf{C}_{\mathcal{G}}(X) = \mathbf{C}_{\mathcal{G}}(Y)$, and assume without loss of generality that $r := \mathrm{rank}(X) \geq \mathrm{rank}(Y)$. Note that $X$ has rank $r \leq 3$, and so it must contain some three points whose rank is also $r$. By applying a permutation to $X$ we can assume without loss of generality that these three points are the first three points. The initial coloring $\mathbf{C_0}(1, 2, 3)(X)$ of this triplet is their Gram matrix $(\langle x_i, x_j \rangle)_{1 \leq i,j \leq 3}$, which has the same rank $r$ as the space spanned by the three points. Next, since $\mathbf{C}_{\mathcal{G}}(X) = \mathbf{C}_{\mathcal{G}}(Y)$ are the same, there exists a triplet of points $i, j, k$ such that $\mathbf{C}_{(1)}(1, 2, 3)(X) = \mathbf{C}_{(1)}(i, j, k)(Y)$ which implies that the initial colorings are also the same. By applying a permutation to $Y$ we can assume without loss of generality that $i = 1, j = 2, k = 3$. Next, since the Gram matrix of $x_1, x_2, x_3$ and $y_1, y_2, y_3$ are identical, there is an orthogonal transformation that takes $x_i$ to $y_i$ for $i = 1, 2, 3$, and by applying this transformation to all points in $X$ we can assume without loss of generality that $x_i = y_i$ for $i = 1, 2, 3$. It remains to show that the rest of the points of $X$ and $Y$ are equal, up to permutation. To see this, first note that $X$ and $Y$ have the same rank since

$$r = \mathrm{rank}(X) \geq \mathrm{rank}(Y) \geq \mathrm{rank}(y_1, y_2, y_3) = \mathrm{rank}(x_1, x_2, x_3) = r.$$

Thus the space spanned by $x_1 = y_1, x_2 = y_2, x_3 = y_3$ contains all points in $X$ and $Y$. Next, we can deduce from the aggregation rule defining $\mathbf{C_1}(1,2,3)(X)$ in (2), that

$$\{\!\{(\langle x_j, x_1\rangle, \langle x_j, x_2\rangle, \langle x_j, x_3\rangle) \mid j \in [n]\}\!\} = \{\!\{(\langle y_j, y_1\rangle, \langle y_j, y_2\rangle, \langle y_j, y_3\rangle) \mid j \in [n]\}\!\}.$$

Since all points in $X$ and $Y$ belong to the span of $x_1 = y_1, x_2 = y_2, x_3 = y_3$, $X$ and $Y$ are the same up to permutation of the last $n-3$ coordinates. This concludes the proof of the theorem. $\qquad\square$

We next outline the completeness proof of the more efficient 2-SEWL.

**Theorem 3.2.** *For every $X, Y \in \mathbb{R}^{3 \times n}$, a single iteration of the 2-SEWL test assigns $X$ and $Y$ the same value if and only if $X \underset{\mathcal{SO}[3,n]}{=} Y$.*

*Proof idea.* The completeness of Vanilla-3-EWL was based on the fact that its initial coloring captures the Gram matrix of triplets of vectors that span the space spanned by $X$, and on the availability of projections onto this basis in the aggregation step defined in (2). Our proof for 2-EWL completeness relies on the fact that a pair of non-degenerate vectors $x_i, x_j$ induces a basis $x_i, x_j, x_i \times x_j$ of $\mathbb{R}^3$. The Gram matrix of this basis can be recovered from the Gram matrix of the first two points $x_i, x_j$, and the projection onto this basis can be obtained from the extra geometric information we added in (18). A full proof is given in the appendix. $\qquad\square$

To conclude this section, we note that the above theorem can be readily used to also show that the 2-EWL test us also complete with respect to $\mathcal{O}[3,n]$. For details see Appendix A.

# 4   WL-equivalent GNNs with continuous features

In the previous section we discussed the generic completeness of 1-EWL and the completeness of 2-SEWL and vanilla 3-EWL. The **Embed** functions in these tests are hash functions, which can be redefined independently for each pair of point clouds $X, Y$. In this section, our goal is to explain how to construct GNNs with equivalent separation power to that of these tests, while choosing continuous, piecewise differentiable **Embed** functions that are injective. While this question is well studied for combinatorial graphs with discrete features Xu et al. [2018], Morris et al. [2018], Maron et al. [2019], Aamand et al. [2022], here we focus on addressing it for Euclidean graphs with continuous features.

## 4.1   Multiset injective functions

Let us first review some known results on injective multiset functions. Recall that a function defined on multisets with $n$ elements coming from some alphabet $\Omega \subseteq \mathbb{R}^D$ can be identified with a permutation invariant function defined on $\Omega^n$. A multiset function is injective if and only if its corresponding function on $\Omega^n$ is separating with respect to the action of the permutation group (see Definition 1.1).

In Corso et al. [2020], Wagstaff et al. [2022] it was shown that for any separating, permutation invariant mappings from $\mathbb{R}^n$ to $\mathbb{R}^K$, the embedding dimension $K$ will be at least $n$. Two famous examples of continuous functions that achieve this bound are

$$\Psi_{sort}(x_1, \ldots, x_n) = \text{sort}(x_1, \ldots, x_n) \quad \text{and} \quad \Psi_{pow}(x_1, \ldots, x_n) = \left(\sum_{i=1}^{n} x_i^t\right)_{t=1}^{n}. \tag{5}$$

When the multiset elements are in $\mathbb{R}^D$, the picture is similar: if there exists a continuous, permutation invariant and separating mapping from $\mathbb{R}^{D \times n}$ to $\mathbb{R}^K$, then necessarily $K \geq n \cdot D$ Joshi et al. [2022]. In Dym and Gortler [2023] it is shown that continuous separating invariants for $D > 1$, with near-optimal dimension, can be derived from the $D = 1$ separating invariants $\Psi = \Psi_{pow}$ or $\Psi = \Psi_{sort}$, by considering random invariants of the form

$$\textbf{Embed}_\theta(x_1, \ldots, x_n) = \langle b_j, \ \Psi\left(a_j^T x_1 \ldots, a_j^T x_n\right)\rangle, \ j = 1, \ldots, K. \tag{6}$$

where each $a_j$ and $b_j$ are $d$ and $n$ dimensional random vectors, and we denote $\theta = (a_1, \ldots, a_K, b_1, \ldots, b_K) \in \mathbb{R}^{K(D+n)}$. When $K = 2nD + 1$, for almost any choice of $\theta$, the function $\textbf{Embed}_\theta$ will be separating on $\mathbb{R}^{D \times n}$. Thus the embedding dimension in this construction is optimal up to a multiplicative constant of two.

An important property of this results of Dym and Gortler [2023] for our discussion, is that the embedding dimension $K$ can be reduced if the domain of interest is a non-linear subset of $\mathbb{R}^{D \times n}$ of low dimension. For example, if the domain of interest is a finite union of lines in $\mathbb{R}^{D \times n}$, then the *instrinsic dimension* of the domain is 1, and so we will only need an embedding dimension of $K = 2 \cdot 1 + 1 = 3$. Thus, the required embedding dimension depends on the intrinsic dimension of the domain rather than on its *ambient dimension*, which in our case is $n \cdot D$.

To formulate these results precisely we will need to introduce some real algebraic geometry terminology (see Basu et al. [2006] for more details): A *semi-algebraic subset* of a real finite-dimensional vector space is a finite union of subsets that are defined by polynomial equality and inequality constraints. For example, polygons, hyperplanes, spheres, and finite unions of these sets, are all semi-algebraic sets. A semi-algebraic set is always a finite union of manifolds, and its dimension is the maximal dimension of the manifolds in this union. Using these notions, we can now state the 'intrinsic version' of the results in Dym and Gortler [2023]:

**Theorem 4.1** (Dym and Gortler [2023]). *Let $\mathcal{X}$ be an $S_n$-invariant semi-algebraic subset of $\mathbb{R}^{D \times n}$ of dimension $D_{\mathcal{X}}$. Denote $K = 2D_{\mathcal{X}} + 1$. Then for Lebesgue almost every $\theta \in \mathbb{R}^{K(D+n)}$ the mapping $\mathbf{Embed}_\theta : \mathcal{X} \to \mathbb{R}^K$ is $S_n$ invariant and separating.*

### 4.2 Multiset injective functions for GNNs

We now return to discuss GNNs and explain the importance of the distinction between the intrinsic and ambient dimensions in our context. Suppose we are given $n$ initial features $(h_1^{(0)}, \ldots, h_n^{(0)})$ in $\mathbb{R}^d$, and for simplicity let us assume they are recursively refined via the simple aggregation rule:

$$h_i^{(t+1)} = \mathbf{Embed}^{(t)} \left( \{\!\{ h_j^{(t)} \}\!\}_{j=1, j \neq i}^n \right). \tag{7}$$

Let us assume that each $\mathbf{Embed}^{(t)}$ is injective on the space of all multisets with $n-1$ elements in the ambient space of $h_j^{(t)}$. Then the injectivity of $\mathbf{Embed}^{(1)}$ implies that $h_i^{(1)}$ is of dimension at least $(n-1) \cdot d$. The requirement that $\mathbf{Embed}^{(2)}$ is injective on a mult-set of $n-1$ features in $\mathbb{R}^{(n-1) \cdot d}$ implies that $h_i^{(2)}$ will be of dimension at least $(n-1)^2 \cdot d$. Continuing recursively with this argument we obtain an estimate of $\sim (n-1)^T d$ for the dimensions of each $h_i^{(T)}$ after $T$ iterations of (7).

Fortunately the analysis presented above is overly pessimistic, because it focused only on the *ambient dimension*. Let us denote the matrix containing all $n$ features at time $t$ by $H^{(t)}$. Then $H^{(t)} = F_t(H^{(0)})$, where $F_t$ is the concatenation of all $\mathbf{Embed}^{(t')}$ functions from all previous time-steps. Thus $H^{(t)}$ resides in the set $F_t(\mathbb{R}^{d \times n})$. Here we again rely on results from algebraic geometry: if $F_t$ is a composition of piecewise linear and polynomial mappings, then it is a semi-algebraic mapping, which means that $F_t(H^{(0)})$ will be a semi-algebraic set of dimension $\dim(\mathbb{R}^{n \times d}) = n \cdot d$. This point will be explained in more detail in the proof of Theo-

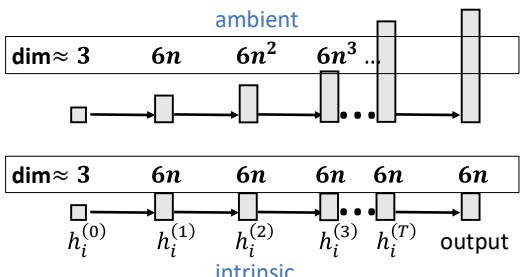

Figure 2: The exponential growth in the dimension that would result from only considering the ambient feature dimension can be avoided by exploiting the constant intrinsic dimension.

rem 4.2. By Theorem 4.1 we can then use $\mathbf{Embed}_\theta$ as a multiset injective function on $\mathcal{X}_t$ with a fixed embedding dimension of $2n \cdot d + 1$ which does not depend on $T$. This is visualized in Figure 2.

**2-SEWLnet** Based on the discussion above, we can devise architectures that simulate the various tests discussed in this paper and have reasonable feature dimensions throughout the construction, In particular, we can simulate $T$ iterations of the 2-SEWL test by replacing all $\mathbf{Embed}^{(t)}$ functions[1] with $\mathbf{Embed}_\theta^{(t)}$, where in our implementation we choose $\Psi = \Psi_{sort}$ in (6). The embedding dimension for all $t$ is taken to be $6n + 1$, since the input is in $\mathbb{R}^{3 \times n}$. We denote the obtained parametric function by $F_\phi$. Based on a formalization of the discussion above, we prove in the appendix that $F_\phi$ has the separation power of the complete 2-SEWL test, and therefore $F_\phi$ is separating.

**Theorem 4.2.** *Let $F_\phi$ denote the parametric function simulating the 2-SEWL test. Then for Lebesgue almost every $\phi$ the function $F_\phi : \mathbb{R}^{3 \times n} \to \mathbb{R}^{6n+1}$ is separating with respect to the action of $\mathcal{SO}[3, n]$.*

To conclude this subsection, we note that while sort-based permutation invariants are used as aggregators in GNNs Zhang et al. [2020, 2018], Blondel et al. [2020], the polynomial-based aggregators $\Psi_{pow}$ are not as common. To a certain extent, one can use the approach in Xu et al. [2018], Maron et al. [2019], replace the polynomials in $\Psi_{pow}$ by MLPs, and justify this by the universal approximation power of MLPs. A limitation of this approach is that it only guarantees separation at the limit.

## 5 Synthetic Experiments

In this section we implement 2-SEWLnet, described in Section 4, and empirically evaluate its separation power, and the separation power of alternative $\mathcal{SO}[3, n]$ invariant point cloud architectures. We trained the architectures on permuted and rotated variations of highly-challenging point-cloud pairs, and measured separation by the test classification accuracy. We considered three pairs of point clouds (Hard1-Hard3) from Pozdnyakov et al. [2020]. These pairs were designed to be challenging for distance-based invariant methods. However, our analysis reveals that they are in fact separable by two iterations of the 1-EWL test. We then consider a pair of point clouds from Pozdnyakov and Ceriotti [2022] which was proven to be indstinguishable by the 1-EWL tests. The results of this experiment are given in Table 1. Further details on the experimental setup appear in Appendix B.

| Separation | complete | $\cong$1-EWL | unknown | unknown | unknown |
|---|---|---|---|---|---|
| Point Clouds | 2-SEWLnet | EGNN | MACE | TFN | GVPGNN |
| Hard1 | 100 % | 100 % | 100 % | 100 % | 100 % |
| Hard2 | 100 % | 100 % | 100 % | 100 % | 50 % |
| Hard3 | 100 % | 100 % | 100 % | 100 % | 95.0 ± 15.0 % |
| Harder | 100 % | 50 % | 100 % | 100 % | 53.7 ± 13.1 % |

Table 1: Separation accuracy on challenging 3D point clouds. Hard examples correspond to point clouds which cannot be distinguished by a single 1-EWL iteration but can be distinguished by two iterations, according to Theorem 2.1. The Harder example is a point cloud not distinguishable by 1-EWL Pozdnyakov and Ceriotti [2022]. GNN implementations and code pipeline based on Joshi et al. [2022].

As expected, we find that 2-SEWLnet, which has complete separation power, succeeded in perfectly separating all examples. We also found that EGNN Victor Garcia Satorras [2021], which is essentially an implementation of 1-EWL, does not separate the **Harder** example, but *does* separate the **Hard** example after two iterations, as predicted by Theorem 2.1. We also considered three additional invariant point cloud models whose separation power is not as well understood. We find that MACE Batatia et al. [2022] and TFN Thomas et al. [2018] achieve perfect separation, (when applying them with at least 3-order correlations and three-order $SO(3)$ representations). The third GVPGNN Jing et al. [2021] architecture attains mixed results. We note that we cannot necessarily deduce from our empirical results that MACE and TFN are complete. While it is true that TFN is complete when considering arbitrarily high order representations Dym and Maron [2020], it is not clear whether order three representation suffices for complete separation. We conjecture that this is not the case. However, finding counterexamples is a challenging problem we leave for future work.

**Future Work** In this work, we presented several invariant tests for point clouds that are provably complete, and have presented and implemented 2-SEWL-net which simulates the complete 2-SEWL test. Currently, this is a basic implementation that only serves to corroborate our theoretical results. A practically useful implementation requires addressing several challenges, including dealing with point clouds of different sizes, the non-trivial $\sim n^4$ complexity of computing even the relatively efficient 2-SEWL-net, and finding learning tasks where complete separation leads to gains in performance. We are actively researching these directions and hope this paper will inspire others to do the same.

---

[1]A minor technicality is that the **Embed** functions are actually defined on vector-multiset pairs. This issue is discussed in the proof of the theorem.

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
