# OpenReview forum: "Complete Neural Networks for Complete Euclidean Graphs"
_NeurIPS.cc/2023/Conference — Submitted to NeurIPS 2023_

### Official Review · Reviewer_7AxF · 2023-06-11

**Soundness:** 3 good
**Presentation:** 2 fair
**Contribution:** 2 fair
**Rating:** 4
**Confidence:** 4

**Summary:**

The authors provided theoretical analyses and proof that the 3-WL algorithm and the Euclidean version of the 2-WL algorithm can distinguish any complete Euclidean graph pairs. The authors then demonstrated that the algorithm can be approximated with GNNs and ran the proposed model on synthetic data to show that it was indeed able to "separate" the hard graph pairs.

**Strengths:**

1. The authors provided rigid mathematical formulation and proof. The problem was well-defined and formulated in the paper.
2. The organization of the paper is clear. The authors first discussed the problem, followed by theoretical analyses and proof. Model architectures and experiments are then provided to support the theoretical claims.
3. The theoretical results are general and can be potentially applied to a wide range of models.

**Weaknesses:**

1. The experiments are inadequate. The authors only tested their model and the baselines on small-scale synthetic datasets. The dataset used in the paper only contains small molecular graph pairs whereas, in practical 3D point cloud scenarios, there can be easily thousands of points. Furthermore, at such a scale, many traditional GNN-based networks including MACE and TFN are also "separating" according to the results. The authors may test their model and the baselines on larger practical datasets.

2. The potential benefit of a model being "separating" was also not experimented. Eventually, we want models to output some meaningful values in the classification or regression task. The author may experiment with such tasks to demonstrate the capability of graph isomorphism tests indeed help with representation learning.

3. The potential application of the algorithm is greatly limited by the assumption that the graph is complete. The authors also mentioned in the future work section that the proposed model scales as $O(n^4)$ with respect to the number of nodes, which is prohibitively large even for small point clouds.

4. The writing of the manuscript needs to be improved in some places. All references were not in parenthesis, making it hard to read the manuscript (e.g. line 143 when quoting the WL test). The quotation marks are not paired. The sections are also a bit strange. I would suggest making the related work and future work a separate section and the current Sec 3 into a subsection (as it is also a theoretical analysis).

**Questions:**

1. Regarding the experiment part, how will the proposed model behave when scaling up to larger practical datasets? (See Weakness 1)

2. Regarding the experiment part, will the capability of graph isomorphism tests help with representation learning and other downstream tasks? (See Weakness 2)

3. The proposed model scales as $O(n^4)$ with respect to the number of nodes. How can it scale up to practical point clouds with thousands of points? Can we drop or relax the *complete graph* assumption? This problem is crucial as the locality assumption is crucial for GNNs.

4. Will the theoretical results hold if locality is assumed? That is, each vertex only aggregates information from a subset of the vertices (its nearby neighbors). I doubt they would also hold under the locality assumption. Consider the simple case of two regular triangles and one regular hexagon, any rounds of 1-EWL will not be able to distinguish from them.

**Limitations:**

The authors have mentioned the limitation of this work in the manuscript and no further negative societal impact is expected.

---

> ### Author Rebuttal · Authors · 2023-08-06
>
> We thank the reviewer for the valuable feedback. Below are our responses to the questions and concerns.
>
> **Concern Regarding Experiments**: As stated in the Author Rebuttal and our responses to the other reviewers, this work is a theoretical work, which does not aim to devise a practical implementation that can compete with state-of-the-art methods. Rather, we proved several novel results regarding the separation power of WL tests. Our synthetic experiments were only meant to empirically validate our theoretical results, and we believe that they are the most suitable tool for this end.
>
> **Benefits of Separation:** It is already established in the literature that separation is a desired property for learning algorithms that respect data symmetries. As mentioned in our response to Reviewer eAUn, it was shown in works such as [Pozdnyakov et al.] [Chen et al.] that methods that lack separation power often have inferior practical performance.
>
> While MACE and TFN succeeded to distinguish the point-cloud pairs in our synthetic experiment, these methods are not guaranteed to separate all pairs of 3D point clouds. We do note, however, that the instances of MACE and TFN evaluated in our experiments consider 3-tuples of points, and thus are likely to have more separation power than MPNNs, which only consider pairs of points. This is further indicated by the superior performance of MACE and TFN over MPNN in our experiments, as well as in real-world tasks [Batatia et al. 2022]. This further correlates with the observation that algorithms with higher separation power tend to have better performance in practical tasks.
>
>
> **Regarding the $\mathcal{O}(n^4)$ running-time complexity of our algorithm:** Our method is the most computationally efficient method to date that provably separates 3D point clouds. Any algorithm on point clouds that is continuous and provably separating is likely to come at a cost of a high running time. While we are not aware of any established lower bounds, the best previous results to date require using prohibitively high-order tensors ($\mathcal{O}(n^{poly(n)})$) to achieve separation [Dym, ICLR 2021], [Lim, ICLR 2023].
>
> We wish to point out the following computational bottleck: Achieving a running time of less than $\mathcal{O}(n^4)$ with an algorithm based on 2-WL, would require developing a continuous and separating embedding of multisets in $\mathbb{R}^3$ that has a running time of less than $\mathcal{O}(n^2)$, which is nontrivial.
>
> We note that once one is willing to forego guaranteed separation, there exist many heuristics to implement WL-based architectures with a significantly reduced running time. Such heuristics include considering local neighbourhoods rather than the whole graph [Feng et al.].
>
> We intend to add a clarification of this issue to the manuscript.
>
> **Regarding the small number of points in the clouds in our evaluation:** Any architecture that is as powerful as high-order k-WL is likely to incur a high computational cost. For example, the well-known PPGN [Maron et al.] requires a running time of $\mathcal{O}(n\^5)$. For practical applications they used a relaxation with a running time of $\mathcal{O}(n\^3)$, which is still inapplicable in settings with thousands of points. Yet, their architecture is considered a cornerstone in the study of learning algorithms on graphs and point clouds.
>
> **Question regarding local neighbourhoods**: Our theoretical guarantees are only applicable to point clouds and to complete Euclidean graphs. The reviewer's understanding is indeed correct that if locality is allowed, then the separation results will not hold.
>
> Relaxations such as allowing locality may nevertheless be used in practice to significantly reduce the running time. We believe that this modification may incur a lesser degradation of performance than using architectures that are based on $1$-WL tests, which are non-separating to begin with. This was demonstrated in [Feng et al.], whose relaxation of a $2$-WL variant to local neighborhoods performed significantly better than MPNNs.
>
> Lastly, we note that $k$-WL tests with $k>1$ are not defined with a notion of locality, and yet they are the predominant method used to upper-bound the separation power of GNNs [Geerts et al.], [Morris et al.], [Maron et al.], some of which do perform aggregation on local neighbourhoods. Thus, the question of separation of graphs assuming full connectivity is still relevant to these models.
>
> **Comments regarding writing:** We highly value the reviewer’s suggestions regarding improving the readability of our manuscript, which we will implement adequately.
>
> **References**
>
> [Geerts et al.] Geerts, F., & Reutter, J. L. (2022). _Expressiveness and approximation properties of graph neural networks. arXiv_ preprint arXiv:2204.04661.
>
> [Chen et al.] Chen, Z., Villar, S., Chen, L., & Bruna, J. (2019). On the equivalence between graph isomorphism testing and function approximation with gnns. _Advances in neural information processing systems_ , 32.
>
> [Morris et al.] Morris, C., Ritzert, M., Fey, M., Hamilton, W. L., Lenssen, J. E., Rattan, G., and Grohe, M. Weisfeiler and leman go neural: Higher-order graph neural networks. In _Proceedings of the AAAI conference on artificial intelligence_, volume 33, pp. 4602–4609, 2019b.
>
> [Maron et al.] Maron, H., Ben-Hamu, H., Serviansky, H., and Lipman, Y. Provably powerful graph networks. _Advances in neural information processing systems_ , 32, 2019.
>
> [Feng et al.] Feng, Jiarui, et al. _Towards Arbitrarily Expressive GNNs in $ O (n^ 2) $ Space by Rethinking Folklore Weisfeiler-Lehman. arXiv_ preprint arXiv:2306.03266 (2023).

---

> > ### Comment · Reviewer_7AxF · 2023-08-11
> > **Comment on Authors' Rebuttal**
> >
> > I appreciate your comprehensive rebuttal regarding my previous questions and concerns. Though some questions are properly addressed, I'd like to reiterate the concerns that I believe did not get well-addressed in the rebuttal.
> >
> > 1.  **The experiments**. I acknowledge the theoretical nature of this work, but still, I don't think the experiments have demonstrated the claimed advantage over other baselines. I do not doubt the completeness of the proposed model, as you have provided solid mathematical proof. Nonetheless, it seems that neither your established theorems nor the experiment results have ruled out the possibility that normal GNNs with locality assumptions also have the expressiveness of distinguishing the geometrics.
> >
> >     You may refer to the GWL paper (https://openreview.net/forum?id=Rkxj1GXn9_) as an example, in which the authors also tried to deal with a theoretical formulation of distinguishability of geometric GNNs and provided experiment results on a wide range of synthetic data. The experiment results from GWL provided evidence for the theory, and the authors also made plausible interpretations of the failed cases, both of which are lacking in this paper.
> >
> > 2. **The motivation/benefit of separation**. I will make a clearer statement why the motivation for separation is somewhat dubious in my view. I'd like to first point out that the separation problem you and Pozdnyakov et al. referred to arises only for non-continuous target functions like categorical information, as previous work (e.g., https://openreview.net/pdf?id=6NFBvWlRXaG) has already demonstrated the universality of TFN for approximating any continuous equivariant functions. For categorical labels, I personally do not consider it necessary to exact Euclidean graph isomorphism. For example, a slightly morphed bunny point cloud should be still recognized as a bunny, and one molecule may adopt various plausible conformations (3D geometries). In these scenarios, we instead want the model to produce somewhat *invariant* predictions to demonstrate robustness to the perturbation or noise on the data. I'd appreciate it if you can come up with some practical applications in which the separation of non-continuous target functions is desired.

---

> > > ### Author Response · Authors · 2023-08-15
> > >
> > > Thank you for your prompt feedback on our rebuttal. We apologize that we have not yet addressed all of your concerns. Below is our response, which we believe should answer both of your remaining concerns.
> > >
> > > ### Concern No. 1 ###
> > >
> > > **Regarding the separation power of normal GNNs:** For most rotation-permutation invariant architectures, the separation power is still unknown. All that is known is that: (a) 1-WL-like architectures do not separate [Pozdynakov et al.]; (2) TFN is universal/separating [Dym-Maron, ICLR 2021]; and now, using our paper, we know that (c) 3-WL- and 2-WL-like architectures are separating. Note that resolving the separation question for each of these three architectures required a non-trivial theoretical paper. Thus, while we agree that it would be interesting to study the separation power of other architectures, this is outside the scope of this paper.
> > >
> > > Specifically, regarding GNNs that allow for incomplete graphs — such architectures will clearly not be complete if, for example, the graph is disconnected.
> > >
> > > **Regarding empirical evaluation:** While we appreciate the extensive empirical evaluation in the GWL paper, note that many of the other papers quoted in our rebuttal (e.g. [Dym-Maron, ICLR 2021], [Villar et al. NeurIPS 2021]) have a much more limited empirical evaluation. For example, [Aamand et al. NeurIPS 22] have only shown simulations of their proposed WL method on sampled graphs.
> > >
> > > Nonetheless, to address your concern, we have further conducted a separation experiment on real-world water tetramer pairs, proposed in [Pozdnyakov et al.]; see results in the following Official Comment. The results demonstrate that our SEWLnet, as well as MACE, achieve separation, while the 1-EWL simulation, TFN and GVPGNN  do not achieve separation. Note that while MACE was able to distinguish the molecule pairs in this experiment, it was not proven to be separating.
> > >
> > >
> > > ### Concern No. 2 ###
> > >
> > > **Regarding Separation and Universality:** Firstly, separation and universal approximation in the continuous domain are tightly interrelated: A model is separating if and only if it can be made universal by composing it from the left with an MLP. For more on this see, e.g. Theorems 17-23 in the GWL paper, Theorem A.1 in our appendix, and the other references discussed in our introduction (the paragraph beginning at line 19). See also Section 2.2 and the figure therein, which illustrate why the failure of separation leads to non-universal models. We will be happy to make the motivation for separation clearer in the main text.
> > >
> > >
> > >
> > > **If we already know that TFN is universal, why bother discussing other algorithms?** Note that the proof of TFN universality requires arbitrarily high-dimensional representations of SO(3), and thus requires the prohibitively high $\mathcal{O}(n^{poly(n)})$-time complexity to achieve universality. In contrast, our result requires only $\mathcal{O}(n^4)$ time. This is accomplished by showing, for the first time, that separation can be achieved using only simple low-dimensional invariants of SO(3): inner products and vector products. The ability to achieve separation/universality using low dimensional invariants was previously recognized as an important open question in several papers, including the paper that proved the universality of TFN [Dym-Maron, ICLR 2021] (see quotes in our Response to All Reviewers).
> > >
> > > In addition, the study of the separation of geometric k-WL algorithms is a natural question in its one right, motivated by the centrality of k-WL algorithms and their separation power in the GNN literature [Geerts et al.].
> > >
> > >
> > >
> > > **On the importance of separation in discrete classification:** Since most discrete classification methods rely on continuous feature vectors, the requirement that these vectors be computed using an invariant, separating, and continuous architecture is very natural:
> > >
> > > - **Invariance:** Guarantees that permutations and rotations do not influence the classification outcome.
> > > - **Separation:** Guarantees that the representation vectors calculated by the model can maintain all the information in the raw input required for classification. Non-separating models may fail in classification due to loss of information.
> > > - **Continuity:** Enables the classifier to be robust to local perturbations.
> > >
> > > We stress that we do not view discrete classification and continuous regression tasks as essentially different, since in practice classification models typically map data to a distribution over the labels using a continuous function that approximates a non-continuous one (e.g. the softmax function vs. 1-hot vector). Such continuous invariant mappings can provably be approximated by separating architectures.

---

> > > > ### Author Response · Authors · 2023-08-15
> > > >
> > > > ### Experiment results ###
> > > >
> > > > | Separation | complete | $\cong$ 1-EWL | unknown | unknown | unknown
> > > > | :---------------- | :------: | ----: | ----: | ----: | ----: |
> > > > | Architecture        |  2-SEWLnet | EGNN | MACE | TFN | GVPGNN |
> > > > | Water Tetramer           |   100 % | 50 % | 100 % | 50 % | 50 % |
> > > >
> > > > ### References ###
> > > >
> > > > [Aamand et al. NeurIPS 22] Aamand, et al. "Exponentially improving the complexity of simulating the Weisfeiler-Lehman test with graph neural networks." Advances in Neural Information Processing Systems 35 (2022).
> > > >
> > > > [Dym-Maron, ICLR 2021] Nadav Dym and Haggai Maron. “On the Universality of Rotation Equivariant Point Cloud Networks” International Conference on Learning Representations (ICLR), 2021
> > > >
> > > > [Villar et al., NeurIPS 2021] Villar, Soledad, et al. "Scalars are universal: Equivariant machine learning, structured like classical physics." Advances in Neural Information Processing Systems 34 (2021).

---

> > > > > ### Comment · Reviewer_7AxF · 2023-08-19
> > > > >
> > > > > I appreciate your detailed follow-up comment regarding my concerns. The new experiments seem more intriguing. However, the following problems remain.
> > > > >
> > > > > I believe the authors did not explicitly address my concerns about the necessity of enforcing strict Euclidean isomorphism on categorical labels. Theoretical guarantees are good, but it seems that, in practical tasks. robustness towards the noises introduced during data curation and even numerical errors introduced by on-the-fly transformations are more important and desirable. If the authors want to demonstrate the importance of geometries, continuous tasks such as molecular dynamics where the geometries play a crucial role would be more suitable instead of graph-level classification.
> > > > >
> > > > > In terms of the universality of TFN, I believe the authors are partially correct in their claim that TFN requires higher degrees of spherical harmonics. In Dym's paper, they demonstrated data with symmetries up to $n$-fold can be captured using the spherical harmonics of $n-1$ degree. This is probably why the TFN failed the water tetramer case -- the latter has a 4-fold symmetry but (vanilla) TFN only uses the degrees up to 1. In practice, point clouds rarely have rotational axes with an order higher than 4; in crystallography, it is well established that rotational symmetries of a regular crystal are limited to 2-fold, 3-fold, 4-fold, and 6-fold. Therefore, given that the time complexity of TFN is $O(L+1)^6$ where $L=n-1$, it provides a more efficient way of achieving completeness than the proposed method.
> > > > >
> > > > > Based on the above observation, I retain my original scores.

---

> > > > > > ### Author Response · Authors · 2023-08-20
> > > > > >
> > > > > > We appreciate your interest in the question of universality and separation. However, we disagree that separation is not relevant for discrete classification. We wish to firstly elaborate on separation in the context of classification, and on its relation to robustness to perturbations.
> > > > > >
> > > > > > Separation guarantees that an architecture is able to distinguish between non-equivalent point clouds. This is a natural requirement for classification tasks. In particular, **lack of separation power may impair classification performance.** For any non-separating $f:\mathbb{R}^{3\times n} \to \mathbb{R}^m$ there exists a low-dimensional point-cloud manifold on which $f$ is constant. In principle, such a manifold may span wide distances, well beyond local perturbations, and possibly across different classes. Therefore, a non-separating architecture might assign the same label to point clouds belonging to different classes.
> > > > > >
> > > > > > On the other hand, we would like to stress that **separating embeddings do _not_ enforce strict Euclidean isomorphism on categorical labels.** They only guarantee that all information in the raw input point cloud is retained for further processing. How this information is dealt with, is up to the architecture that follows the embedding.
> > > > > >
> > > > > > This brings us to our final point: **A continuous separating embedding, composed with a universal approximator, _promotes_ perturbation robustness.** If $f$ is a continuous separating embedding, and $X \in \mathbb{R}^{3\times n}$ is a point cloud, then small changes to $X$ can only cause small changes to $f(X)$. Composing $f$ with a universal approximator $g$ yields a predictor $g \circ f$ that, upon successful training, can learn to be resilient to perturbations that do not change an object's class. This way, a slightly morphed bunny will still be recognized as a bunny.
> > > > > >
> > > > > > **Regarding our classification experiment:** Such experiments are standard practice in similar theoretical works, such as GWL and [Pozdnyakov et al.]. The latter reference also shows that ***stronger separation in these experiments leads to stronger performance on a real-world water tetramer chemical regression task*** [Pozdnyakov et al. Fig. 8] (in their experiment they document continuous loss rather than classification error rate, but the experiment is indeed categorical). Similarly, in the context of graphs, separation capabilities lead to strong real-world results [Chen et al.].
> > > > > >
> > > > > > **Regarding TFN:** We reiterate that almost-everywhere completeness is not the same as completeness. As we mentioned in this manuscript, 1-WL-equivalent architectures separate almost all point clouds, yet are not complete, thus we cannot approximate all continuous functions on point clouds using these models. The same is true for TFN. This has practical implications as mentioned in the above paragraph.
> > > > > >
> > > > > > Additionally, we are not aware that [Dym-Maron 2021] have shown that TFN can separate point clouds with $n$-fold symmetries with the complexity you state. The running-time complexity is based on expressing high-order polynomials, and TFN does _not_ attain universality with the complexity you stated.
> > > > > >
> > > > > > Could you possibly be referring to the GWL paper [Joshi et al.]? If so, please note that they experimented with identifying the _orientation_ of structures with rotation symmetry higher than $n$-fold. This does _not_ imply the separation of point clouds with such symmetries.
> > > > > >
> > > > > > **References:**
> > > > > >
> > > > > > [Dym-Maron 2021] Nadav Dym and Haggai Maron. “On the Universality of Rotation Equivariant Point Cloud Networks” International Conference on Learning Representations (ICLR), 2021
> > > > > >
> > > > > > [Joshi et al.] Joshi, C. K., Bodnar, C., Mathis, S. V., Cohen, T., & Lio, P. (2023). On the expressive power of geometric graph neural networks. arXiv preprint arXiv:2301.09308.
> > > > > >
> > > > > > [Pozdnyakov et al.] Pozdnyakov, Sergey N., and Michele Ceriotti. "Incompleteness of graph neural networks for points clouds in three dimensions." Machine Learning: Science and Technology 3.4 (2022).
> > > > > >
> > > > > > [Chen et al.] Chen, Z., Villar, S., Chen, L., & Bruna, J. (2019). On the equivalence between graph isomorphism testing and function approximation with gnns. Advances in neural information processing systems , 32.

---

> > > > > > > ### Author Response · Authors · 2023-08-20
> > > > > > >
> > > > > > > We would like to add a general remark. We are sorry to see that we have not been able so far to convince you about the importance of the theoretical study of separation and universality of rotation-invariant architectures. Nonetheless, we feel that it is not justified to disqualify the paper on these grounds, given that there is a considerable community of researchers attending NeurIPS who _are_ interested in these questions. This has been demonstrated extensively in the introduction to the paper, in our response to all reviewers, and in the recent response of Reviewer Z66V.
> > > > > > >
> > > > > > > In light of the above, we believe that the grounds for acceptance or rejection should focus not on whether the problem of separation is deemed interesting, but rather on the extent to which this paper contributes to this problem. We believe that our contribution is substantial, as this paper is the first to establish full separation and universality using low-order representations.

---

> > > > > > ### Comment · Reviewer_Z66V · 2023-08-20
> > > > > > **As a fellow reviewer, please reconsider your score in case there are misunderstandings**
> > > > > >
> > > > > > Dear fellow reviewer,
> > > > > >
> > > > > > I believe there are a few misunderstandings. I would like to invite you to reconsider your position based on the following:
> > > > > >
> > > > > > > I believe the authors did not explicitly address my concerns about the necessity of enforcing strict Euclidean isomorphism on categorical labels. Theoretical guarantees are good, but it seems that, in practical tasks. robustness towards the noises introduced during data curation and even numerical errors introduced by on-the-fly transformations are more important and desirable.
> > > > > >
> > > > > > You could think of isomorphism as a proxy for **injectivity** of the hidden representation/feature vector learnt by the neural network (up to symmetry), i.e. one-to-one mapping of inputs to feature vectors.
> > > > > >
> > > > > > If you can learn to assign a unique feature vector to each unique input, you can subsequently solve any representation learning task, in theory (whether it be classification/regression).
> > > > > >
> > > > > > Additionally, isomorphism/injectivity and theoretical guarantees around them help us start building the **mathematical foundations** towards tackling even harder and (perhaps) more practical questions regarding generalisation and optimisation.
> > > > > >
> > > > > > In summary, injectivity/isomorphism/theory of expressive power helps us understand what is the **'best case scenario'** for our neural nets. Yes, its not perfectly practical because training real neural nets involves understanding generalisation and optimisation, but it has proven useful in advancing neural network architecture design.
> > > > > >
> > > > > > >  In Dym's paper, they demonstrated data with symmetries up to n-fold can be captured using the spherical harmonics of n-1 degree.
> > > > > >
> > > > > > Having revisited the paper by [Dym and Maron (ICLR 2021)](https://openreview.net/forum?id=6NFBvWlRXaG), I believe this is a misunderstanding. There are no mentions regarding rotational symmetry fold in the paper (I assume by n-fold, you mean n-fold rotationally symmetric point clouds).
> > > > > >
> > > > > > The Dym-Maron paper proves that, in order for TFN to be universal, **one requires tensor rank/dimension to be infinite**. In fact, the paper's conclusion ends with "On the other hand, an interesting open problem is understanding whether universality can be achieved using only low-dimensional representations". This is the precise question answered by the present paper!
> > > > > >
> > > > > > I personally found [this follow up paper by the same authors](https://proceedings.mlr.press/v196/finkelshtein22a/finkelshtein22a.pdf) to be a more approachable version of the original Dym-Maron paper. For instance, Section 3 discusses clearly the universality properties of TFN and TFN-style cartesian tensor models on Euclidean point clouds.
> > > > > >
> > > > > > There was a key research gap post the story started by Dym-Maron, and this paper did a good job filling it, in my opinion.
> > > > > >
> > > > > > > In practice, point clouds rarely have rotational axes with an order higher than 4; in crystallography, it is well established that rotational symmetries of a regular crystal are limited to 2-fold, 3-fold, 4-fold, and 6-fold.
> > > > > >
> > > > > > I am guessing: you meant that the GWL paper (Joshi et al., ICML 2023) showed the connection between rotationally symmetric arrangements of point clouds and the tensor rank/dimension.
> > > > > >
> > > > > > I believe that particular experiment was regarding injectivity of a single layer of TFN-like models. On the other hand, Dym-Maron is about universality/injectivity at the model level.
> > > > > >
> > > > > > > Based on the above observation, I retain my original scores.
> > > > > >
> > > > > > As a fellow reviewer, I'd like to request you to reconsider. There is a large community interested in this line of work. **I do not think the authors should be burdened with justifying the existence of an entire research area.** I believe it is our job as reviewers to carefully check whether this paper is technically correct and novel.
> > > > > >
> > > > > > Best regards,
> > > > > >
> > > > > > Reviewer Z66V

---

> > > > > > > ### Comment · Reviewer_7AxF · 2023-08-20
> > > > > > > **Comment to Reviewer Z66V & Authors**
> > > > > > >
> > > > > > > Dear Reviewer Z66V & Authors,
> > > > > > >
> > > > > > > I would like first to express my appreciation again for your detailed responses. I would also like to apologize for one confusion I have caused in the previous follow-up comments. As Z66V has carefully pointed out, I referred to the GWL for two related papers: **Dym and Maron (ICLR 2021)** which demonstrated the completeness of geometric GNNs, and **Joshi et al. (PMLR 2023)** which empirically evaluated TFN on highly symmetrical data. The latter paper is available on [OpenReview](https://openreview.net/forum?id=Rkxj1GXn9_) (the link in my initial review) or on [ArXiv](https://arxiv.org/pdf/2301.09308.pdf). The empirical claim that TFN with spherical harmonics of degree $L$ can handle the $(L+1)$-fold symmetry came from the latter paper. In this sense, I believe my previous concerns regarding the practical comparison with TFN still hold. Nonetheless, given the late publish date of the latter paper and the theoretical nature of this work, I will **raise my score from 3 to 4**. I will then justify why I will not give higher scores.
> > > > > > >
> > > > > > > I noticed that there seems to be a misunderstanding of my major concerns. I am concerned with **the practical application and necessity of strict Euclidean isomorphism**. I totally agree with Reviewer Z66V and the authors' claim on the theoretical rigor and significance. However, my concerns regarding the practical scenarios await responses. To simplify our discussion, I request the authors to address the following question in an ML scenario on geometric representation learning:
> > > > > > >
> > > > > > > > Assume that due to numerical issues, the distance between a point pair is enlarged by $10^{-5}$ unit after applying some rotation. Should the original point cloud be distinguished from the rotated one?
> > > > > > >
> > > > > > > You may notice the above example provides a kind of extreme case, but in practice, multiple factors like noise during data curation and numerical errors make it more desirable to achieve **robustness** instead of **completeness** (or "separatingness"). The high computational complexity and the practical evaluation of TFN are other (but minor) points that hamper the practical application of the proposed model.
> > > > > > >
> > > > > > > Again, I'd like to point out that my previous comments are **not final** and I am still **open to discussion**. I believe I have adequately provided justification regarding my evaluation of this work so far.
> > > > > > >
> > > > > > >
> > > > > > > Regards, Reviewer 7AxF.

---

> > > > > > > > ### Author Response · Authors · 2023-08-20
> > > > > > > >
> > > > > > > > We thank you for raising the score, and for your willingness to continue and invest your time in engaging in discussion regarding our paper. We apologize that our previous explanation did not convey our message. Perhaps indeed it will be helpful to concretely address the scenario you suggested:
> > > > > > > >
> > > > > > > >
> > > > > > > > > Assume that due to numerical issues, the distance between a point pair is enlarged by $10^{-5}$ unit after applying some rotation. Should the original point cloud be distinguished from the rotated one?
> > > > > > > >
> > > > > > > >
> > > > > > > > A typical approach for handling such a classification task is to use an invariant point-cloud embedding $f:\mathbb{R}^{3 \times n} \to \mathbb{R}^m$ composed with a fully connected network whose last layer is a softmax; namely, $g:\mathbb{R}^m \to \Delta(C)$, where $\Delta(C) \subset [0,1]^{C}$ is the probability simplex and $C$ is the number of classes. Since $g \circ f$ is continuous, small perturbations of $X$ should only cause small perturbations of $g \left( f \left(X\right)\right)$. Thus, if $g \circ f$ assigns label $c$ to $X$ with high confidence/probability, then this assignment should remain essentially the same under a small perturbation.
> > > > > > > >
> > > > > > > > We stress that **separation does not contradict robustness**, rather these are two independent important properties. Generally speaking robustness to noise is related to the continuity of the model. Separation is important, since models that lack separation power may fail, by construction, in classification tasks where non-separable point clouds belong to different classes.
> > > > > > > >
> > > > > > > >
> > > > > > > > To  further address your concern regarding noise, we conducted the following separation experiment:
> > > > > > > >
> > > > > > > > We picked the 5 highly symmetrical point cloud pairs from our experiment in the paper. The point clouds in each pair are not isomorphic to one another. For each pair (A, B) we gave the first point cloud a distinct label, A, and a distinct label to the second point cloud, B. Each point cloud pair was replicated 1000 times and we added independent Gaussian noise with mean zero and variance of 0.1 to each replica and additionally permuted and rotated each replica. We then trained the model on classifying this dataset. (This experiment is analogous to the experiment in the manuscript but with added noise).
> > > > > > > >
> > > > > > > > We find that ***our simulation of 2-SEWL indeed still performs with perfect accuracy***, see table below. 1-EWLsim, which denotes a simulation of the 1-EWL test suggested by [Podznyakov], does not separate the 1-WL indistinguishable synthetic and real-life counterexamples, Harder and Water Tetramer, respectively. Thus our classification results reported in the paper were not affected by the addition of noise.
> > > > > > > >
> > > > > > > > Furthermore, we note that simulations of variations of high order WL, for instance, $N^2$-GNN [Feng et al.], has attained significantly improved performance compared to less separating 1-WL-equivalent models, such as MPNN [Wu et al.], on the QM9 chemical regression task. ***Based on the analysis in this work, this model is separating (it is as expressive as 3-WL), “exacts” a simulation of high-order WL, and is robust to noise in real-life datasets such as QM9.***
> > > > > > > >
> > > > > > > >
> > > > > > > > **Regarding the ”claim that TFN with spherical harmonics of degree L can handle the (L+1)-fold symmetry”:** We believe there is still a misunderstanding regarding the experiment results in [Joshi et al.]. [Joshi et al., Table 2] claim that layers using order L tensors are unable to identify the orientation (rotation) of structures with rotation symmetry higher than  L-fold. This does _not_ imply order L TFN can separate point clouds with L-1-fold symmetries. There is no claim in [Joshi et al.] or in [Dym-Maron] that TFN is complete on such structures or that it has lower time complexity than our method. We reiterate that TFN requires $\mathcal{O}(n^{poly(n)})$ complexity for separation/universality, while we require $\mathcal{O}(n^4)$.
> > > > > > > >
> > > > > > > > In general, separating models will *always* separate non-equivalent point clouds. It is very difficult to know based on a handful of examples whether TFN or other architectures are "separating in practice", since TFN may be "called into duty" in a variety of different tasks and molecules.
> > > > > > > >
> > > > > > > > **Tables:**
> > > > > > > >
> > > > > > > > | Architecture   | 2-SEWLnet |  1-EWLsim
> > > > > > > > |----------------------|-----------|------------|
> > > > > > > > | Hard 1 (noise)        | 100%      | 100%
> > > > > > > > | Hard 2 (noise)        | 100%      | 100%
> > > > > > > > | Hard 3 (noise)        | 100%      | 100%
> > > > > > > > | Harder (noise)        | 100%      | 50%
> > > > > > > > | Water Tetramer (noise)| 100%      | 50%
> > > > > > > >
> > > > > > > > | Architecture   | 2-SEWLnet |  1-EWLsim
> > > > > > > > |----------------|-----------|------------|
> > > > > > > > | Hard 1       | 100%      | 100%
> > > > > > > > | Hard 2        | 100%      | 100%
> > > > > > > > | Hard 3       | 100%      | 100%
> > > > > > > > | Harder       | 100%      | 50%
> > > > > > > > | Water Tetramer | 100%      | 50%
> > > > > > > >
> > > > > > > > **References:**
> > > > > > > >
> > > > > > > > [Feng et al.] Feng, Jiarui, et al. Towards Arbitrarily Expressive GNNs in
> > > > > > > >  Space by Rethinking Folklore Weisfeiler-Lehman. arXiv preprint arXiv:2306.03266 (2023).

---

### Official Review · Reviewer_eAUn · 2023-07-07

**Soundness:** 4 excellent
**Presentation:** 3 good
**Contribution:** 2 fair
**Rating:** 5
**Confidence:** 4

**Summary:**

The paper analyzes neural networks for point clouds toward modeling of geometric phenomena. It considers the application of message passing networks/GNNs to Euclidean graphs, whereby a variation of the well-studied k-WL test is adapted to point clouds by using a complete graph on the point cloud and making use of Euclidean pairwise distances. To this effect, the authors propose the k-EWL test and show that: (1) For k=1, two iterations of message passing are sufficient to separate most point clouds in any dimension, (2) A single iteration is sufficient for all point 3D point clouds when k=3. Furthermore, additional differential architectures are proposed and demonstrated to have similar separation power as k-EWL tests.

I think the paper has some promise and is generally well-written. But it can be strengthened by better motivating the problem and providing more detailed experimentation, ideally on chemistry/molecular datasets (given that this was cited as a motivation/application in the intro). I think the paper needs some more work, but addressing some of these points would make me open to raising my score.

**Strengths:**

(1.) The paper is generally easy to follow and well-written. I understood the definitions and theorem statements without any problem.
(2.) GNNs + point clouds seems like an underexplored area, and the authors make progress in this area (motivation nonwithstanding).

**Weaknesses:**

(1.) I think there is some motivation lacking for why one wishes to separate point clouds via GNNs, or why it is desirable to construct variants of k-WL. The paper hints at the importance of this for chemical applications, but there isn't much discussion about this beyond the intro.
(2.) The formulation of EWL doesn't seem novel; it is simply a standard MPNN on a complete graph with use of distance as an edge feature in the update rule (in the framework of Gilmer et. al 2017). SEWL seems more interesting, but the motivation is a bit lacking.
(3.) The experiments could be stronger. While the authors provide experiments on synthetic datasets of point clouds demonstrating effectiveness of the proposed architectures, the paper could benefit from some experiments on real-world chemistry tasks, as chemical applications, biological molecular datasets, etc. were cited as a motivation in the intro.

**Questions:**

Suggestions:

(1.) Make the motivations in the intro stronger (see weakness above).
(2.) Including further experiments, e.g., on real world chemistry datasets (in line with the motivation in the intro) would highly strengthen the paper.

Questions:

(1.) Could the authors comment on non-neural net based approaches for the task? How does the proposed approach compare to these? The methods in Table 1 seem to be all NN-based.

**Limitations:**

Some limitations are highlighted in the Future Work section at the end of the paper.

---

> ### Author Rebuttal · Authors · 2023-08-05
>
> We thank the reviewer very much for the valuable feedback. Below are our responses to the questions and concerns.
>
> **Motivation for Separation**: Separation is a desired property for machine-learning algorithms on point clouds, which bears both practical and theoretical importance. For example, it can be shown that for any neural network that is not separating, there exists a continuous invariant function that it cannot approximate. Thus, non-separating neural networks are not universal approximators of invariant functions.
>
> This theoretical weakness often manifests as inability to reach a low training loss in real-world tasks. For instance, [Pozdnyakov et al.] have shown that a model that does not separate two point clouds that are non-isomorphic, has poorer performance in chemical regression task; see [Pozdnyakov et al. Section v Figure 8]. Similarly, GNNs that cannot separate non-isomorphic graphs show inferior empirical performance in practical learning tasks compared to GNNs that do separate [Chen et al.].
>
> **Motivation for WL:** The Weisfeiler-Leman (WL) hierarchy is currently the predominant method of measuring the separation capability of GNNs. Notable examples include [Morris et al., Xu et al., Maron et al. 2019b]. Architectures such as these are widely used for real-world tasks on point clouds [Feng et al., Maron et al. 2019b].
>
> New variants of high-order WL are frequently introduced to obtain more robust or fine-grained separation power. Here we introduced a novel variant of 2-WL to obtain provable separation of point clouds with a lower time complexity than existing methods.
>
> We will modify our manuscript to clarify this motivation.
>
> **Concern Regarding Experiments:**
> As mentioned in our general response above, this paper is a theoretical paper, whose aim is not to propose a specific algorithm to compete with state of the art, but rather to analyze the separation power of the k-WL test on point clouds. The main purpose of the included experiments was to validate our theoretical results. We believe that our synthetic experiments are sufficient for this purpose.
>
> Nonwithstanding, our theoretical results have high relevance to architectures that are used in practice, as many models that are as powerful as 3-WL show a strong performance on real-world tasks, e.g. the QM9 molecular property prediction benchmark [Maron et al, Feng et al.]. Furthermore, our proposed 2-EWL lays the foundation for the development of architectures with a strong separation capability and a lower running time than the aforementioned 3-WL-based methods.
>
> **Regarding the novelty of EWL tests:** While 1-EWL has been previously proposed [Pozdnyakov et al.], $k$-EWL tests with k>1, as well as $k$-SEWL tests, are indeed a novelty in our work.
>
> Moreover, 1-EWL based on MPNN, introduced in [Pozdnyakov et al.], is a template algorithm, which relies on black-box embeddings of multisets in $\mathbb{R}$ that are required to be separating. Another novelty of our work is that we introduce a concrete $\textit{instantiation}$ of this test, with a continuous separating embedding. This endows our instantiation the power to separate almost any 3D point cloud, while maintaining efficient running time.
>
> The original motivation for developing the SEWL tests was to be invariant to rotations but not to arbitrary orthogonal transformations, e.g. reflections. We then used this test to derive the 2-EWL test, which is separating on 3D point clouds while having a lower computational and memory complexity than the vanilla 3-EWL. We then propose a continuous implementation of this test in $O(n^4 \cdot log(n))$ time — the lowest complexity of a continuous separating algorithm that we are aware of.
>
> **Non-Neural Network Methods:** We thank the reviewer for this suggestion. We intend to add a discussion of non-neural methods such as [Bigi et al.], [Drautz], [Dusson et al.] to our manuscript and consider evaluating some of them in our synthetic experiments.
>
> **References:**
>
> [Bigi et al.] Bigi, Filippo et al. _“Wigner kernels: body-ordered equivariant machine learning without a basis.” ArXiv_ abs/2303.04124 (2023): n. pág.
>
> [Drautz] Ralf Drautz, “Atomic cluster expansion for accurate and transferable interatomic potentials,” Phys. Rev. B 99, 014104 (2019).
>
>  [Dusson et al.] Genevieve Dusson, Markus Bachmayr, G´abor Cs´anyi, Ralf Drautz, Simon Etter, Cas van der Oord, and Christoph Ortner, “Atomic cluster expansion: Completeness, efficiency and stability,” Journal of Computational Physics 454, 110946 (2022).
>
> [Gasteiger et al. ] Gasteiger, Johannes, Florian Becker, and Stephan Günnemann. "Gemnet: Universal directional graph neural networks for molecules." Advances in Neural Information Processing Systems 34 (2021): 6790-6802.
>
> [Dym et al. 2023] Dym, Nadav, and Steven J. Gortler. "Low dimensional invariant embeddings for universal geometric learning." arXiv preprint arXiv:2205.02956 (2022).
>
> [Zhao et al. NeurIPS 2022] Zhao, Lingxiao, Neil Shah, and Leman Akoglu. "A practical, progressively-expressive GNN." Advances in Neural Information Processing Systems 35 (2022): 34106-34120.
> (Please note that due to insufficient space, further references are in the Author Rebuttal )

---

> > ### Comment · Reviewer_eAUn · 2023-08-21
> > **Thank you for the response**
> >
> > Thank you for clarifying several points and taking the time to answer my questions. Also, I'm happy to see that you will add a discussion on non-neural network based methods to the write-up.
> >
> > I'm raising my score accordingly.

---

> ### Comment · Reviewer_Z66V · 2023-08-17
> **Response by another reviewer**
>
> Dear fellow reviewer,
>
> > I think there is some motivation lacking for why one wishes to separate point clouds via GNNs, or why it is desirable to construct variants of k-WL. The paper hints at the importance of this for chemical applications, but there isn't much discussion about this beyond the intro.
>
> I'd like to point you to an emerging line of work on separating point clouds via GNNs, both from a theoretical and experimental perspective. There is a growing interest in separation as a design principle for 3D geometric GNNs -- models' relative abilities to separate point clouds is one possible measure of **expressive power**.
> - PhysRev: https://arxiv.org/abs/2001.11696
> - NeurIPS: https://arxiv.org/abs/2206.07697
> - ICML: https://arxiv.org/abs/2301.09308
>
> (These are just some prominent/recent ones. These models are being used for a myriad of AI for Science applications.)
>
> This paper and others on separation give us a useful mental framework to compare architectures in this emerging and important class of models in an abstract manner, while removing implementation details.
>
> > The formulation of EWL doesn't seem novel
>
> I believe the authors don't claim it is novel, either. They build upon the work of Pozdynakov-Ceriotti-2022 (https://iopscience.iop.org/article/10.1088/2632-2153/aca1f8/meta).
>
> Best,
>
> Reviewer Z66V

---

### Official Review · Reviewer_Z66V · 2023-07-08

**Soundness:** 3 good
**Presentation:** 3 good
**Contribution:** 3 good
**Rating:** 7
**Confidence:** 5

**Summary:**

This paper studies the theoretical completeness of neural networks for Euclidean/3D point clouds, from the perspective of whether they can distinguish all non-isomorphic point clouds.

Key theoretical contributions include showing that variations of the k-WL graph isomorphism test are complete for 3D point clouds, and that distance-based 1-WL tests are complete for *almost all* point clouds (measure theoretic perspective).

The work also demonstrates that a GNN can be designed with the proposed completeness guarantees, and sanity checks the theoretical results on synthetic counterexamples from previous studies.

**Strengths:**

- This work shows that adaptations of the k-WL hierarchy of graph isomorphism tests can be 'complete' on 3D point clouds. I believe this is a **novel** theoretical contributions for neural networks on point clouds in Euclidean space.

- I believe the findings are **significant**, as neural networks on Euclidean graphs and point clouds are an emerging area of interest from both theoretical and applied perspectives.

- The paper is **well written** and **clear** in terms of presentation:
    - The Introduction does a good job highlighting the research gap.
    - The coverage of related work in Section 1.1 is useful.
    - Section 2 makes a good bridge from WL to the Euclidean setting.

- I went through the proofs, which are correct to the best of my understanding.

**Weaknesses:**

- It seems challenging to translate this paper's ideas into practice as the model's parameters depend on the number of points $n$ taken as input. This probably makes it very difficult to build a trainable model **while retaining** theoretical guarantees.
    - The authors are upfront about this when discussing limitations.

- Beyond sanity-checking the theoretical ideas on the counterexample from Pozdnyakov-Ceriotti, 2022, the synthetic experiment does not seem to provide any further insights into practical instantiations of the ideas in this paper, or about this class of models more broadly.



**Questions:**

Questions and clarifications:

- Regarding E(n) Equivariant GNN (Satorras et al.) being less expressive than 1-EWL: won't more than one iteration of E(n) Equivariant GNN be able to distinguish between the counterexample of Pozdnyakov-Ceriotti, 2022?
    - The appendix states that the experiments use the QM9 variant of E(n) Equivariant GNN without the position updates. If you meant that this version is less expressive than 1-EWL: yes, in that case I agree, but that version of the model is not what I as a reader would usually consider E(n) Equivariant GNN. That model is invariant.

- Regarding the relationship between Theorem 3.2, Figure 2, and 2-SEWLnet: technically, if a layer is injective or complete (as you prove for one iteration of 2-SEWL), **why do we even need to stack multiple of them?**

- Regarding Theorem 2.1 and Theorem A.2:
    - The main takeaway here is that 1-EWL is sufficient to separate almost all point clouds. The reason is that, for counterexamples which cannot be separated by 1-EWL, the size of the manifold that those counterexamples belong to is very small w.r.t. that of all possible point clouds. Is my understanding correct?
    - On line 533-535, I tried to follow the argumentation but: (1) Could you expand on why the dimensionality is <= 3n-1? (2) Basu et al. is a textbook; is there a better reference? Is this a very simple result?

- Regarding experimental setup: Why were different # of layers used for different models?

Suggestions:

- In Figure 1, it may be useful to draw the actual graphs and also state the exact figures from the Pozdnyakov papers that each graph is taken from, to be least ambiguous.
- Fix the citation for the E(n)-GNN paper (it should appear as Satorras et al., 2021 and include the rest of the authors).
- Consider adding the equation for 2-EWL after eq.4.
- Consider discussing the ACE framework and MACE in Related Work, as this is a framework for building a complete basis for equivariant functions on a set of points up to some interaction body order.
- Typo in line 235.


**Limitations:**

The authors have adequately addressed the limitations but not discussed any potential negative social impact.

Beyond what the authors mention regarding practical instantiation of their models, one major theoretical limitation is that the framework is restricted to complete geometric graphs, and the construction of complete/universal models for the general sparse graph setting remains an open question. This my be worth reiterating.

---

> ### Author Rebuttal · Authors · 2023-08-05
>
> Thank you very much for your valuable feedback. Below are our responses to your questions/concerns.
>
> **Concerns regarding translating ideas into practice:** Indeed the separation guarantee incurs a high computational cost. However, once one is willing to forego guaranteed separation, there exist many heuristics to implement WL-based architectures with a significantly reduced running time, while having minimal impact on the performance in real-life tasks; for example [Feng et al.] [Morris et al.].
>
> **Concern regarding experiments:** This manuscript aims to theoretically prove the expressiveness of WL-based architectures on point clouds — a fundamental theoretical gap not addressed in the literature. Developing an efficient implementation of the WL hierarchy for practical tasks is indeed an interesting further research direction. However, it is outside the scope of this paper.
>
> **Regarding EGNN:** Indeed our choice of naming was inaccurate and has misattributed the lack of separation to the original EGNN, while the non-separating algorithmm used in our experiments was the variant with no coordinate updates. We will change the model name from EGNN to 1-EWLsim, denoting a simulation of 1-EWL, to avoid this lack of clarity.
>
> **On Stacking Multiple Layers:** While indeed one layer of our network is injective, and thus it can be used to approximate any continuous function (see Lemma A.1), it is often required to stack multiple leayers to learn high-level representations. This was observed in a variety of architectures in many domains; see [Bengio] for a detailed discussion of this phenomenon. We will add a clarification on this to the manuscript.
>
> **Regarding Theorem 2.1 and Theorem A.2:** (a) Yes, the set of counterexamples that cannot be distinguished by 1-EWL is indeed of measure zero. (b) To prove this, we show that this set is contained in the zero-set of a nontrivial polynomial. We then combine this with the fact that the zero-set of a non-trivial polynomial is always dimension deficient, and thus has measure zero. This is a well-known result, which holds all for the larger class of analytic functions as well. See, for example, Proposition 3 in [Mityagin 2020].
>
> **Number of layers:** The different number of layers is based on the default choice of hyperparameters of the respective architectures, which we attempted to optimize. We presented the results with the best choice of hyperparameters for each architecture.
>
> **Suggestions:** We highly value the reviewer’s suggestions and shall incorporate all of them into our manuscript.
>
> **References**
>
> [Pozdnyakov et al.] Pozdnyakov, S. N., & Ceriotti, M. (2022). _Incompleteness of graph convolutional neural networks for points clouds in three dimensions. arXiv_ e-prints, arXiv-2201.
>
> [Bartók et al.] Bartók, Albert P., Risi Kondor, and Gábor Csányi. "On representing chemical environments." Physical Review B 87.18 (2013): 184115.
>
> [Caron, Richard] Caron, Richard. (2005). _The Zero Set of a Polynomial._ 10.13140/RG.2.1.4432.8169.
>
> [Bengio] Bengio, Yoshua. "Learning deep architectures for AI." _Foundations and trends® in Machine Learning 2.1_ (2009): 1-127.
>
> [Cybenko] Cybenko, George. "Approximation by superpositions of a sigmoidal function." _Mathematics of control, signals and systems_ 2.4 (1989): 303-314.
>
> [Chen, Chi et al.] Chen, Chi et al. “Graph Networks as a Universal Machine Learning Framework for Molecules and Crystals.” _Chemistry of Materials_ (2018).
>
> [Schutt et al.] Schütt, Kristof T., et al. "Schnet–a deep learning architecture for molecules and materials." _The Journal of Chemical Physics_ 148.24 (2018).
>
> [Xu et al.] Xu, Keyulu, et al. _"How powerful are graph neural networks?." arXiv_ preprint arXiv:1810.00826 (2018).

---

> > ### Comment · Reviewer_Z66V · 2023-08-16
> > **Concerns addressed; score increased**
> >
> > Thank you for the clarifications. My concerns have been addressed, and I have raised my score to reflect this.
> >
> > > Developing an efficient implementation of the WL hierarchy for practical tasks is indeed an interesting further research direction. However, it is outside the scope of this paper.
> >
> > I agree. I only meant that the model which realizes this paper's theory, **in its current form**, may not be practical. But I agree that this is not the paper's main contribution. This could even be a direction for follow up work.
> >
> > > We will change the model name from EGNN to 1-EWLsim
> >
> > That makes sense. I think one would expect an equivariant model when reading 'EGNN'.
> >
> > > While indeed one layer of our network is injective, and thus it can be used to approximate any continuous function (see Lemma A.1), it is often required to stack multiple leayers to learn high-level representations.
> >
> > Understood. This aspect of the paper is very interesting!
> >
> > > Mityagin 2020
> >
> > Perhaps you missed adding this reference? I cannot find a single author paper by Mityagin in 2020 (https://scholar.google.com/citations?hl=en&user=Yyaun24AAAAJ&view_op=list_works&sortby=pubdate).

---

> > > ### Author Response · Authors · 2023-08-17
> > >
> > > We highly appreciate your interest in this proof and we apologize for accidentally omitting the reference for it.
> > >
> > > **Reference:** The dimension reduction argument is stated in Proposition 2 in [Mityagin 2020].
> > >
> > > [Mityagin 2020] Mityagin, B.S. The Zero Set of a Real Analytic Function. Math Notes 107, 529–530 (2020). https://doi.org/10.1134/S0001434620030189
> > >
> > > **Context regarding Proposition 2:**
> > >
> > > Proposition 2 states the Hausedorff dimension of the zero set of a non-zero analytic function is deficient. The Hausedorff dimension of a manifold is always greater or equal to its topological dimension (the dimension of the manifold), see Theorem 6.3.10 in [Edgar 2007], thus this reduction applies to the dimension of the manifold we defined in Equation 11, Theorem A.2 in our manuscript.
> > >
> > >
> > > **References:**
> > >
> > > [Edgar 2007] Edgar G. Measure, Topology, and Fractal Geometry. Undergraduate Texts in Mathematics. New York: Springer; 2007.
> > >
> > > [Mityagin 2020] Mityagin, B.S. The Zero Set of a Real Analytic Function. Math Notes 107, 529–530 (2020). https://doi.org/10.1134/S0001434620030189

---

### Official Review · Reviewer_5HoU · 2023-07-10

**Soundness:** 3 good
**Presentation:** 2 fair
**Contribution:** 3 good
**Rating:** 5
**Confidence:** 2

**Summary:**

This paper seeks to theoretically demonstrate the complete determination of point clouds, up to permutation and rigid motion. The authors formulate a Euclidean variant of the 2-WL test, effectively illustrating the separation capacity of the Euclidean Graph Neural Network on highly symmetrical point clouds.

**Strengths:**

1. The paper delivers a theoretical exploration of point cloud completeness.
2. It discusses the separation capability of the Euclidean Graph Neural Network in high-dimensional representations.

**Weaknesses:**

1. In appendix Line 564, what does $(\star)$ stand for?
2. Does the proposed method risk confounding reflection equivariance?

**Questions:**

See above

---

> ### Author Rebuttal · Authors · 2023-08-06
>
> We thank the reviewer for the feedback. Below are our responses to your questions/concerns.
>
> **Syntax:** $(\star)$ stands for “they both have rank r, and $x_i \neq x_j$”. Then we refer to this fact in the proceeding sentence.
>
> To improve clarity, we will replace  (*) with “Due to the fact that they both have rank r, and $x_i \neq x_j$,”. We thank the reviewer for this point.
>
> **Regarding Reflection Equivariance:** Our proposed method can accommodate both rotation invariance as well as simultaneous rotation and reflection invariance. In particular, the SEWL test is invariant to rotations but not reflections, and the EWL test is invariant to both rotations and reflections. Please let us know if we adequately addressed your question.
>
> On another note, [Villar et al.] characterize rotation equivariant functions via rotation invariant functions. Thus, results in this manuscript may be used to obtain rotation equivariant universal models. This is an active direction for future work.
>
> [Villar et al.] Villar, Soledad, et al. "Scalars are universal: Equivariant machine learning, structured like classical physics." Advances in Neural Information Processing Systems 34 (2021): 28848-28863.

---

### Author Rebuttal · Authors · 2023-08-07

**Response to All Reviewers**

We would like to thank the reviewers for their helpful remarks and detailed feedback, which we have read carefully. We were glad to see the reviewers recognized our novel theoretical contribution. Yet, we feel that we did not convey the significance of our results in the context of separation and $k$-WL tests. Let us attempt to remedy this:

Separation is a desired property for machine-learning algorithms, which bears both theoretical and practical importance. For example, neural networks that cannot separate point clouds are provably $\textit{not}$ universal approximators of continuous point-cloud functions. This theoretical weakness may lead to hinderence of performance in real-world tasks such as molecular property prediction [Pozdnyakov et al. 2022] and regression on social-network graphs [Chen et al.].

Many recent popular architectures for point clouds are based on $k$-WL tests [Morris et al.], [Feng et al.], [Maron et al.]. Yet, prior to our work, no such architecture was proven to separate point clouds. In this work we show for the first time that any architecture that is as expressive as 3-WL is provably separating. In addition, we propose a novel variant of 2-WL that is also provably separating, while being more computationally efficient than 3-WL.

Regarding our experiments, the main focus of our work is theoretical. As such, its aim is not to propose a practical algorithm, but rather to answer a long-standing theoretical question regarding the separation capability of WL-based architectures. For this purpose we used synthetic experiments, as we believe that they are the most suitable means to validate our theoretical results.

We note that this is common practice, as many theoretical papers on this topic have been published in NeurIPS based on the strength of their theory, e.g. [Joshi et al., ICML 2023], [Villar et al., NeurIPS 2021], [Dym-Maron, ICLR 2021], [Wagstaff, ICML 2019], [Aamand, NeurIPS 2022]. Such theoretical results often proved valuable in the subsequent development of practical methods. For instance, [Dym-Maron, ICLR 2021] has inspired GemNet [Gasteiger et al.], a widely used GNN.

To conclude, we believe that the novel theoretical results presented in this manuscript will be of interest to the research community, and may lay the foundations for further theoretical as well as practical research. Below we provide several quotes from recent papers to establish our claim:

“Proposition 2 states that this architecture [Invariant Graph Network (IGN) applied to Gram matrices] universally approximates O(d) invariant and permutation equivariant functions. The full approximation power requires high order tensors to be used for the IGN; in practice, we restrict the tensor dimensions for efficiency …” [Lim, ICLR 2023]

“... provably universal equivariant frameworks are such in the limit in which they generate high-order correlations… It is an interesting, and open, question whether a given order suffices to guarantee complete resolving power.” [Pozdnyakov, MLST 2022]

“... an interesting open problem is understanding whether universality can be achieved using only low-dimensional representations.” [Dym, ICLR 2021]


**References**

[Aamand, NeurIPS 2022] Aamand, Anders, et al. "Exponentially improving the complexity of simulating the Weisfeiler-Lehman test with graph neural networks." Advances in Neural Information Processing Systems 35 (2022).

[Villar, NeurIPS 2021] Villar, Soledad, et al. "Scalars are universal: Equivariant machine learning, structured like classical physics." Advances in Neural Information Processing Systems 34 (2021).

[Wagstaff, ICML 2019] Wagstaff, Edward, et al. "On the limitations of representing functions on sets." International Conference on Machine Learning. PMLR, 2019.

[Dym, ICLR 2021] Nadav Dym and Haggai Maron. “On the Universality of Rotation Equivariant Point Cloud Networks” International Conference on Learning Representations (ICLR), 2021

[Lim, ICLR 2023] Derek Lim, Joshua Robinson, Lingxiao Zhao, Tess Smidt, Suvrit Sra, Haggai Maron, Stefanie Jegelk “Sign and Basis Invariant Networks for Spectral Graph Representation Learning.” International Conference on Learning Representations (ICLR 2023)

[Pozdnyakov, MLST 2022] Pozdnyakov, Sergey N., and Michele Ceriotti. "Incompleteness of graph neural networks for points clouds in three dimensions." Machine Learning: Science and Technology 3.4 (2022).

[Geerts et al.] Geerts, F., & Reutter, J. L. (2022). _Expressiveness and approximation properties of graph neural networks. arXiv_ preprint arXiv:2204.04661.

[Chen et al.] Chen, Z., Villar, S., Chen, L., & Bruna, J. (2019). On the equivalence between graph isomorphism testing and function approximation with gnns. _Advances in neural information processing systems_ , 32.

[Pozdnyakov et al.] Pozdnyakov, S. N., & Ceriotti, M. (2022). _Incompleteness of graph convolutional neural networks for points clouds in three dimensions. arXiv_ e-prints, arXiv-2201.

[Morris et al.] Morris, C., Lipman, Y., Maron, H., Rieck, B., Kriege, N. M., Grohe, M., Fey, M., and Borgwardt, K. _Weisfeiler and leman go machine learning: The story so far. arXiv_ preprint arXiv:2112.09992, 2021.

[Feng et al.] Feng, Jiarui, et al. _Towards Arbitrarily Expressive GNNs in $ O (n^ 2) $ Space by Rethinking Folklore Weisfeiler-Lehman. arXiv_ preprint arXiv:2306.03266 (2023).

[Gasteiger et al. ] Gasteiger, Johannes, Florian Becker, and Stephan Günnemann. "Gemnet: Universal directional graph neural networks for molecules." Advances in Neural Information Processing Systems 34 (2021).

[Maron et al.] Maron, H., Ben-Hamu, H., Serviansky, H., and Lipman, Y. Provably powerful graph networks. Advances in neural information processing systems , 32, 2019.

---

### Author Response · Authors · 2023-08-18
**Upcoming End of Discussion Phase**

Dear reviewers, thanks again for your detailed reviews. Please let us know if we have adequately addressed your concerns, so we have sufficient time to amply respond before the end of the discussion phase on Monday.

Thanks!

---

### Decision · Program_Chairs · 2023-09-21

**Decision:**

Reject

**Comment:**

The paper studies completeness and universality properties of geometric graph neural networks over point clouds. The reviewers appreciated the theoretical contributions, however there were concerns regarding the practical relevance of the architecture, limited empirical assessment, and trade-offs of the separation property with other useful properties in this context, such as invariance/robustness. I encourage the authors to address and discuss these points further in a future submission.